# Important Influencing and Decision Factors in Organic Food Purchasing in Hungary

**Kinga Nagy-Pércsi [1] and Csaba Fogarassy [2],***

[1]  Institute of Regional Economics and Rural Development, Szent Istvan University, Pater Karoly st. 1, 2100 Godollo, Hungary; nagyne.percsi.kinga@gtk.szie.hu

[2]  Climate Change Economics Research Centre, Szent Istvan University, Pater Karoly 1, 2100 Godollo, Hungary

*   Correspondence: fogarassy.csaba@gtk.szie.hu

**Abstract:** Organic farming is one of the most developed and accepted production systems from the aspect of sustainability. In this study, the Hungarian organic market was segmented on the basis of attitude-relating motivations of organic food purchasing. A descriptive statistic was used for the whole sample, and factor and cluster analysis was applied to segment the organic consumers. A sample of 247 questionnaires was processed to investigate the behavior and characteristics of Hungarian organic food consumers. Our aim was to explore the Hungarian organic consumer market from the aspect of trust in labels, we would like to find answers to the questions "Which kind of information consumers check on the product?" and "What are the main influencing factors of purchase decisions?". According to our research, the majority of organic food consumers say that color is not as important as taste or freshness, so they do not associate the color of the product with the freshness of the product. Based on the factor analysis, four clusters could be separated that show different shopping motives and differ in their attitude towards food shopping. We named these clusters: Health-conscious, Disappointed, Safe and free food eaters, and Quality-price comparators. Based on the recognized consumer segments, different information should be communicated to consumers so that they can use it in their consumer decisions. Our research suggests that branding or product labeling is not as important to organic food consumers as we previously thought.

**Keywords:** organic consumer market; emotional factors; health consciousness; consumers' trust; labeling system; consumer behavior; bio food

---

## 1. Introduction

In the case of organic food marketing the usual direct selling form and through this the personal contact with the producers fulfill the circular economy principle also. Namely, the producers should know the demand and the consumers should know the supply to avoid overproduction or overdemand. In this context, when the organic food product leaves the short supply chain, or direct supply chain an applicable quality certification, a label becomes necessary. However, a well-functioning label is a basic instrument in the development of the sector as well as other aspects also, the realization of the circle of utilized and produced materials is a principle of organic farming also. What are the main influencing factors of purchase decisions? How can consumers' trust in certificating labels increase? To answer these questions the organic food consumers' behavior should be investigated. A key factor for organic agriculture is the perception of consumers related to organic products, in terms of attitude and preferences, as particular expressions of their behavior [1]. When analyzing consumer behavior, one should consider the following: What consumers think/ perceive, feel, and how they behave, alongside environmental factors that influence them [2]. Consumer behavior can hence be influenced by experience. This can lead to a change in attitudes and behavior [3]. The factors that

determine consumer perception refer not only to physical needs, like food, health and environmental protection [4], but also their dependence on other socio-cultural components, like culture, religion, training, income, and social position [5]. The most important incentive for the consumers in Hungary, similar to many other countries eating organic food, is the healthiness of this kind of product [6,7]. Different motives can be separated into a healthy diet. There are special needs as a consequence of illnesses, fitness considerations, search for safety, and avoiding harmful components. Hungarian food consumption habits have recently been influenced by several trends, such as the trend of convenience, health and wellness, environmental consciousness, search for experience, ethical consumption, and time consciousness [8].

The results of the Nielsen Global Health and Wellness Survey [9], conducted in 60 countries and involving 30,000 consumers, revealed that the most desirable food attributes are freshness, naturalness, and minimal processing [10]. Fresh food can fulfill most of the requirements relating to these attributes. On the basis of our survey, organic food consumers eat fruits and vegetables most frequently and the purchasing habits relating to fruits and vegetables are also relevant in the organic food sector. The research institutes GFK Hungary Ltd. and Agrar Europe Ltd. [11] conducted a consumer survey relating to fresh fruit and vegetable consumption and purchasing. According to their results, the consumers buy fresh fruits and vegetables mainly at traditional markets, from small farmers or street vendors today also. According to GFK analysis, these sources gave 27% of the total purchased quantity. The consumers buy mainly fresh food on market, 36% of the total spending went on vegetables, 18% on fruits, 12% on fresh meat, 7-7% on bakery and processed meat products [12]. In spite of the favorable health impact of fresh food, the relating microbiological and chemical risks give a reason for anxiety [13,14]. The illnesses relating mainly to the sporadic cases of microbiological hazards [15–17]. The foodborne diseases connected to the fresh products, unfortunately, are gaining importance in the last few decades. The fresh products form a separate food safety category. The bacterial agents found the most common food safety danger by the expert in relation to fresh food, this was followed by foodborne viruses and pesticide residues. The different mold toxins can have also an important health risk. It is interesting, however, that the organic food consumers judge the microbiological dangers as not so harmful, they think the residues are the most dangerous health affected factor [18]. Other food safety hazards like antibiotic resistance, the wax shield on the fruits shell, and the genetically modified organizations are all emerging problems for the stakeholders in the fresh food supply chain [19–21]. For these anxieties, eating organic food can be a solution for a great part of the consumers.

Parallelly, there is also an increasing interest in investigating the health effects of organic food consumption. However, the results are still insufficient when attempting to formulate explicit conclusions [22]. The abovementioned concerns are influencing the development of the organic markets on the world and have an impact on the consuming pattern and behavior of organic food consumers. These phenomena, the purchasing channels of organic food, the most frequently purchased food in certain purchasing channels, the attitudes of purchase and the relating subjects should be investigated, in particular, for a better understanding of the economic characteristics of this special market. Based on the above, the research questions in this study are: Do consumers in the organic market pay attention to where they buy? Do customers consider it safer to buy food from small-scale farmers? Is the appearance of the product important, or is the taste important, and how do brand and product labels influence customer decisions?

## 2. Literature Review

The relevant literature affects three main fields: The main motives of buying organic, the preferred supply chain in organic food purchasing and the utility and acceptance of labels relating to organic food marketing. However the organic food consumption is at a low level in Hungary, the demand for organic food is growing steadily. In 2010, the market value of organic products was 82.3 million USD, which took 1.5% of the total food trade and meant a great increase as compared to 2005. In this year the total organic selling was only 36.7 million USD, 0.8% of the total food trade. For 2015 the

value of organic food selling was assessed to 110.4 million USD [23,24]. This upward trend can also be observed in other parts of the world and relating mainly to environmental concerns [25]. The healthy diet and lifestyle are also becoming more and more important for the consumers parallelly with economic growth and this process is favorable for the development of organic selling. The analyses of major motivations that stand behind the organic food buying behavior of consumers reveal that health issues represent the main reason for purchasing organic food and that health attributes have become as important as sensory ones during the buying decision-making process [26–29]. The reduced consumption of chemicals in organic farming is the main criterion for which the consumers choose products. When it comes to the respondents' perception of the sensory quality of the organic products, it can be said that a majority of the respondents consider organic products less appealing but instead tastier. The results of a Romanian study show a positive consumer perception for the taste of the organic products, indifferent to the level of education [1]. Consumers' interest in organic foods in Hungary is also driven by the perceived health benefits associated with consuming goods free of chemical additives and pesticide residue [30,31]. On the other hand, solidarity with local producers, and the associated environmental benefits also drive sales. A positive relationship can be found between higher education and organic food acceptance [32,33]. Beside education income situation has a great impact on buying organic. Consumers with higher income buy organics more frequently [33–37]. Women were suggested to be organic food buyers [38–40]. Women are more motivated due to eating a healthy diet, men are more influenced by their social circumstances [41]. The organic food buyers tend to be older, with children, and have a higher education level than those of non-buyers [33,34,37,42,43]. Consumers' urge to seek novelty and to gain substantial information regarding product utility in terms of price and quality can also influence consumers' decisions to buy organic products [44]. According to the beforementioned it can be stated that organic food is strongly motivated by consumers' perception that organic food is healthier than conventional food [45]. This is particularly relevant in emerging markets where healthiness is perceived to be the most important characteristic of organic food that motivates consumer purchase behavior [46]. Sensory and the so-called ethical quality characteristics mentioned by many studies as a motivation factor to buy organics [47]. The organic food buyers have an inclination to pay a higher price for the higher food safety requirements [48]. It should be parallel mention that according to Csíkné [49] the most important influencing factors at food procurement in the case of an average Hungarian consumer are the price, freshness, food-safety, and the choice. She found that the least important influencing factors are the direct personal contact with the farmers, the producing methods and decreasing of the environmental pollution. Hungarian organic production and processing are underdeveloped, in 2009 almost 70% of the organic food was stemming from import. Hungarian organic food stores concentrate on vegan food and only a few of them occupy meat products however it can be more adequate for traditional eating habits [50]. Szente et al. [51] mentioned that in Hungary several times products, which are not in demand being distributed, while the selection and volume of certain products are not satisfactory on organic markets. It is contrary to the principle of circular economy also namely the suppliers should know the demand. This fact also emphasizes the role of a well -functioning label. A well-functioning label can build also trust in organic food marketing. The findings of a Chinese survey revealed that information on the label of organic food is a significant antecedent of consumers' trust in organic food [52]. Rácz found that domestic consumers do not know the objective meaning of food labels in several cases. This uncertainty can be resulted by the number of labels, so consumers cannot gather a wide range of information before purchasing decision because of the lack of unified, sustainability proving labels and the use of several label formats if we take into account also the limiting role of time [53]. The different domestic promotion campaign, also the labeling of those food products which contains raw materials produced in Hungary or those which are produced in Hungary has been operating for years. However, the special marketing program of the organic food produced in Hungary and the connected label system does not work at all. The lack of an adequate label comes out in other countries also. The Romanian consumers do not pay sufficient attention to organic food labels. The authors assumed that it is due to that the Romanians are not

properly educated in this regard and because of hasty shopping [7]. Drexler et al. found that organic product labeling can play a role in decision-making, but regardless 27% of experiment participants do not care about the organic quality labels or don't pay attention to them [54]. Due to the lack of an adequate labeling system, the consumers' trust, the personal and direct consumer-producer connection is the dominant factor in purchasing decisions of organics. The organic food consumers are interested in who has produced food items they consume and where they have originated. Organic food consumers may also be inspired by knowing and supporting the individual who has produced their food as opposed to supporting a faceless corporation or distant producer [30,55]. According to the survey conducted by Szente [40] the origin is partially or totally important for the respondents (72.9%) and those who prefer organic food also pay attention to the local origin. It should be added here that the alternative and modern form of direct selling are not popular yet in Hungary. Most of the consumers rather choose the traditional Short Supply Chains like producers' market and organic markets [48]. Hungarian consumers especially price-sensitive [43]. It should be noted here that the most important limiting factor is the price in the development of the market. Other limiting factors are the availability problems and the lack of trust relating to the labeling systems and certification processes [43]. Makatouni [56] added to these factors the lack of perceived value [57]. This phenomenon is general in other countries with developing the organic market in the world [58]. According to other international studies, the relationship between producer and buyer is also determined by behavior associated with the cultural or solidarity economy, which in combination with a number of other features, may form a different relationship or network within the consumer system [59–61]. Due to the above-mentioned phenomena, it can be stated that organic food consumers are not very interested in branded products, but rather they are looking for product groups or opting for system characteristics related to product sales that are based on consumer confidence. Hungarian organic food consumers are categorized by several authors [8,30,43,55]. The families with small children as a subcategory can be well defined inside the "health-conscious" category. Families with small children are those who give special interest to organic food. Between 2006 and 2010 the selling of organic baby food and the special baby dairy product started to increase. During the financial crisis, the families focused their spending on their children and they choose an organic baby meal which they think safer and healthier as compared to others [62].

## 3. Material and Methods

### 3.1. Sampling and Survey Instrument

To get a deeper insight into the characteristics of organic food marketing and consumption in Hungary a survey was conducted on the biggest Hungarian Organic Market in Budapest (Biokultúra Organic Market) in February 2018. This was the first step in data sampling. There 31 questionnaires were collected altogether by personal interviews. Many useful experiences were gained from this survey and the questionnaire can be improved according to these experiences. The respondents had the opportunity to add their opinion relating to certain questions. This information was noted and used for data processing. Parallelly with these interviews, to find and eliminate potential problems relating to the survey instrument, a pre-test was performed. An evaluation group consisting of three academics experts was formed to ensure the validity and suitability of the items. The applied questionnaire (Appendix A) contained 16 mainly closed questions relating to eating habits, consumer behavior, factors influencing consumers' purchase decisions, attitudes, purchasing channels and judgment of food-safety beyond the demographic characteristics of the respondents. The questionnaire was established mainly on the basis of the relevant professional literature [12,25,28,40,49,51,54] selected according to the aim of the article. Food choice motives were assessed using 18 motive dimensions, like "freshness", "taste", "colour", "wrapping", "advertisement", "impact on health", "components", "price", "high preparedness", "high endurance", "habits", "price", "free of E-numbers", free of additives", "nutrients", "recommendation", "label", "brand". The scale was a five-point Likert

scale which was anchored at "1" indicating strong disagreement and "5" indicating strong agreement. The consumption frequency of certain food products was measured by a scale ranging from 0-5, where 0 means "I do not consume.", 1 means "I consume less than once in a month.", 2 means "I consume once or twice in a month.", 3 means "I consume once or twice in a week.", 4 means "I consume three or four times in a week.", 5 means "I consume on a daily basis.". After the abovementioned on the spot survey further 811 were collected with the help of students attending to the courses of "Food safety and quality assurance" and "Hygiene in catering" in Szent Istvan University. The exercise of the students was to interview one person from their family and from other familiar households (grandmother, grandfather, aunts, uncles, friends, etc.) until the end of March 2018. It was an important criterion that the respondents should be more than 18-year-old. Finally, 842 questionnaires were collected in these ways. From the 842 questionnaires only 247 was suitable to investigate the behavior and characteristics of organic food consumers because together with the personal interviews conducted on the Organic Market in Budapest, 247 respondents buy regularly organic food. Inside this group, another subgroup can be separated called organic market consumers. The most important characteristic of this group, that they buy organic food on some of the organic markets as the most important purchasing channel. The size of this sample is 102 questionnaires, which contains the questionnaires filled in the frame of a personal interview on the biggest Organic Market in Budapest. Unfortunately, 13 questionnaires should be excluded from further analysis because of too many missing data or inconsistency. However, we processed the data of 18 questionnaires where the respondents choose the option that they do not buy organic food but they parallelly use the organic market as a food purchasing channel. The problem could be stemming from the misunderstanding of the adequate concepts because in Hungary the official term for this kind of food is ecological (means organic) but many consumers know them as bio food.

*3.2. Data Analysis*

A descriptive statistic was used for the whole sample and factor and cluster analysis was applied to segment the organic consumers. The data were analyzed using SPSS software, version 24. Factor analysis was performed and segmentation was conducted using K-means cluster analysis. The factor scales consisting of six factors were used in cluster analysis. Before K-means clustering a hierarchical cluster analysis using Ward Linkage was conducted to determine the adequate number of clusters. The results of this cluster analysis indicated that the optimal number of clusters was 4 (Figure 1). The differences between the segments were examined using the average related consumer attitude scores for certain clusters and the average scores of purchasing motives.

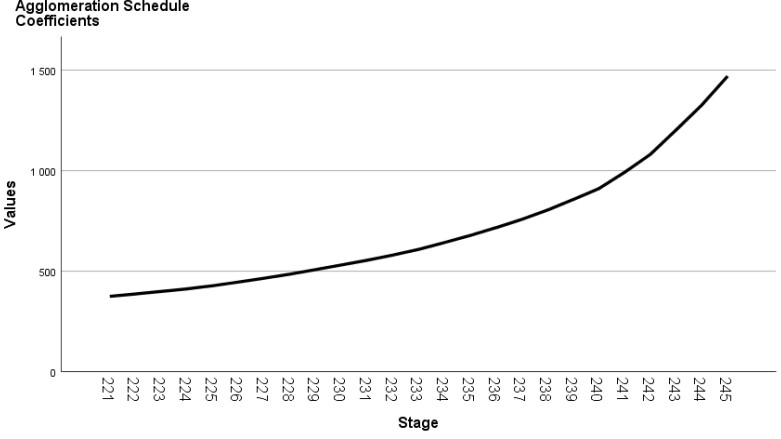

**Figure 1.** Agglomeration schedule of hierarchical cluster analysis.

The last two agglomeration steps were omitted on the bases of the Dendrogram and the elbow criterion method. As a result, the ideal numbers of the clusters were 4. As Table 1 shows we could get homogenous groups by using 4 clusters.

**Table 1.** Most important attributes of the different groups.

| Cluster Number of Case | | REGR Factor Score 1 for Analysis 1 | REGR Factor Score 2 for Analysis 1 | REGR Factor Score 3 for Analysis 1 | REGR Factor Score 4 for Analysis 1 | REGR Factor Score 5 for Analysis 1 | REGR Factor Score 6 for Analysis 1 |
|---|---|---|---|---|---|---|---|
| 1 | Mean | 0.3719936 | 0.0305698 | 0.2100684 | −0.3274677 | −1.2811828 | −0.2956395 |
| | N | 59 | 59 | 59 | 59 | 59 | 59 |
| | Std. Deviation | 0.90315332 | 0.88929638 | 0.79562377 | 0.91567436 | 0.72484566 | 1.12373810 |
| 2 | Mean | −0.1319210 | −0.6791270 | 0.2445502 | 0.5734791 | 0.2775804 | 0.1043713 |
| | N | 88 | 88 | 88 | 88 | 88 | 88 |
| | Std. Deviation | 0.90315332 | 0.69337555 | 0.68250015 | 0.89487064 | 0.65204011 | 1.05759567 |
| 3 | Mean | −0.1519007 | −0.5637794 | −2.2113915 | −0.5323831 | 0.4719697 | 0.0564277 |
| | N | 21 | 21 | 21 | 21 | 21 | 21 |
| | Std. Deviation | 1.16376660 | 0.54495091 | 1.08128650 | 1.19386063 | 0.69237934 | 0.90991976 |
| 4 | Mean | −0.0916494 | 0.8948581 | 0.1605740 | −0.2559682 | 0.5288633 | 0.0906803 |
| | N | 78 | 78 | 78 | 78 | 78 | 78 |
| | Std. Deviation | 1.05559132 | 0.73491381 | 0.64267461 | 0.82070203 | 0.70093961 | 0.81436277 |
| Total | Mean | 0.00000000 | 0.00000000 | 0.00000000 | 0.00000000 | 0.00000000 | 0.00000000 |
| | N | 246 | 246 | 246 | 246 | 246 | 246 |
| | Std. Deviation | 1.00000000 | 1.00000000 | 1.00000000 | 1.00000000 | 1.00000000 | 1.00000000 |

The variables shown in Table 2 were subjected to a factor analysis using principal axis factoring and Varimax rotation to determine the smallest number of meaningful factors.

**Table 2.** Variables used for segmentation.

| Food Choice Motives | Mean | SD |
|---|---|---|
| price | 3.585366 | 1.215054 |
| wrapping | 2.735772 | 1.264629 |
| high-endurance | 3.069106 | 1.218606 |
| advertisement | 2.065041 | 1.126597 |
| origin | 3.626016 | 1.311789 |
| brand | 3.028455 | 1.32911 |
| label/certification | 3.239837 | 1.341412 |
| components | 4.146341 | 1.047313 |
| additives | 4.056911 | 1.012581 |
| high-preparedness | 2.971545 | 1.179417 |
| nutrients | 3.646341 | 1.110675 |
| free of E-number | 3.678862 | 1.228289 |
| taste | 4.373984 | 0.946731 |
| colour | 3.211382 | 1.30776 |
| freshness | 4.544715 | 0.83056 |
| recommendation | 3.170732 | 1.263021 |
| habit | 3.45122 | 1.263159 |
| good impact on health | 4.329268 | 0.926546 |

Bartlett's test of sphericity was significant at the 0.001 level and the Kaiser–Meyer–Olkin (KMO) value was greater than 0.7 [63].

## 4. Results

The results of the study give a clear answer that organic food consumers are not very interested in the appearance of the product, unlike traditional consumers who make their decisions based on the appearance of the product. Our investigations also show us where they are and what kind of shopping environment organic food consumers are looking for. What kind of cultural or solidarity elements of these decisive locations can be based on the analysis!

### 4.1. Main Characteristics Of Organic Food Consumers

Most of the organic food consumers (60%) is female, 45% of the respondents live in Budapest, in the capital city, 51% graduated and 44% is white-collar worker. These data are in harmony with the results of former surveys also [64,65]. In that case, 86% of the surveyed consumer have average or higher income levels. The respondents of the survey think in the first place with the same scores that they are conscious consumers, they pay much attention to where they buy food and that the food is full of harmful ingredients. The respondents agreed in that to a great extent. They do not really think that they get safer food on the market but at the same time, they do not trust food traded by food stores (Figure 2).

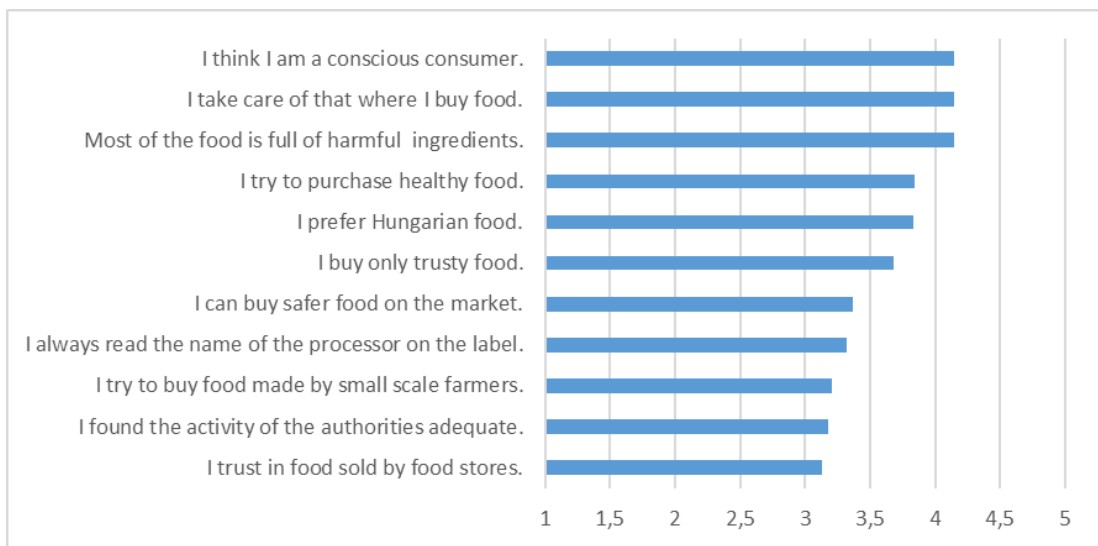

**Figure 2.** The attitude of the surveyed organic food consumers (N = 247).

According to the survey, the most important purchase influencing factors are freshness, flavor, and only at the third place positive health impacts. It is interesting to note here that Oroian et al. [6] found that the "extrinsic attributes" of the organic products were not considered to be the main reasons for consumers to buy organic products, but it was appreciated due to the important information on the ingredients and nutritional aspects, the factors that influence organic food consumers' buying behavior. It is important and on the contrary to the average Hungarian consumers' behavior that the price is not really important for this segment (Figure 3).

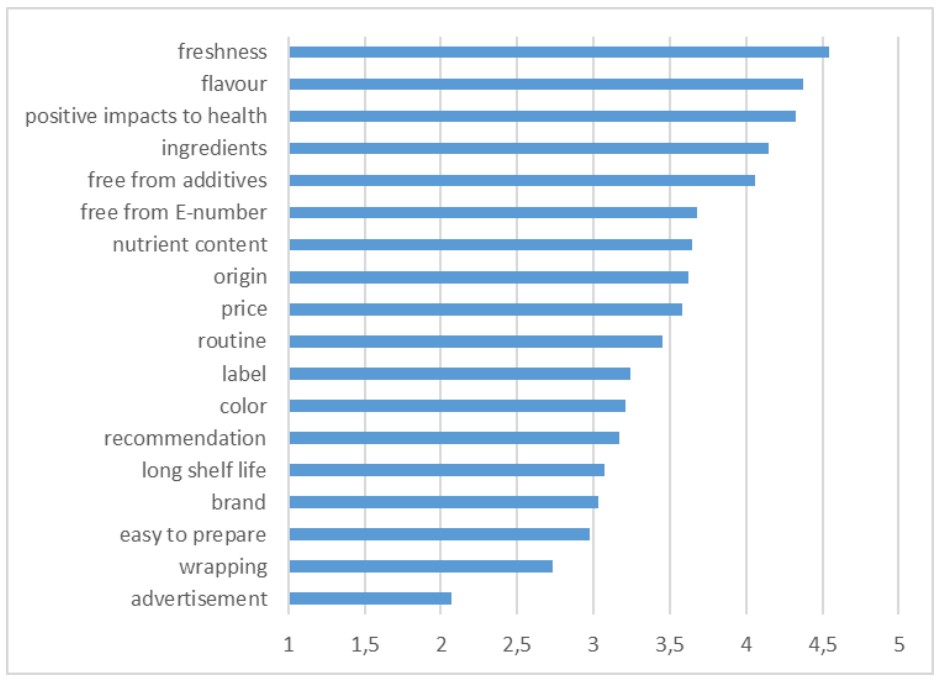

**Figure 3.** The influencing factors of purchase in the case of organic food consumers (N = 247).

These attributes are not surprising in light of the fact that the organic market consumers eat vegetables most frequently according to our survey. Our survey sample contained not only organic food consumers but also non-organic food consumers. Moreover, a segment could be separated from the organic food consumers, those who buy organic food basically from the organic market, these are

the organic market consumers (N = 102). For comparison, the main features of the other two segments from our survey sample are hereby presented in figures. The differences between the diet of the non-organic food eaters and organic market consumers are bigger.

This latter mentioned group eats vegetables and fruits with the highest frequency (Figure 4), while the non-organic food consumers eat fruits and vegetables only at the fifth and fourth place (Figure 5). There is a slight difference between the diet of organic food consumers and organic market consumers (who buy organics mainly on organic markets) because the organic food consumers eat vegetables with the highest frequency. It was followed by the bakery and only then fruits.

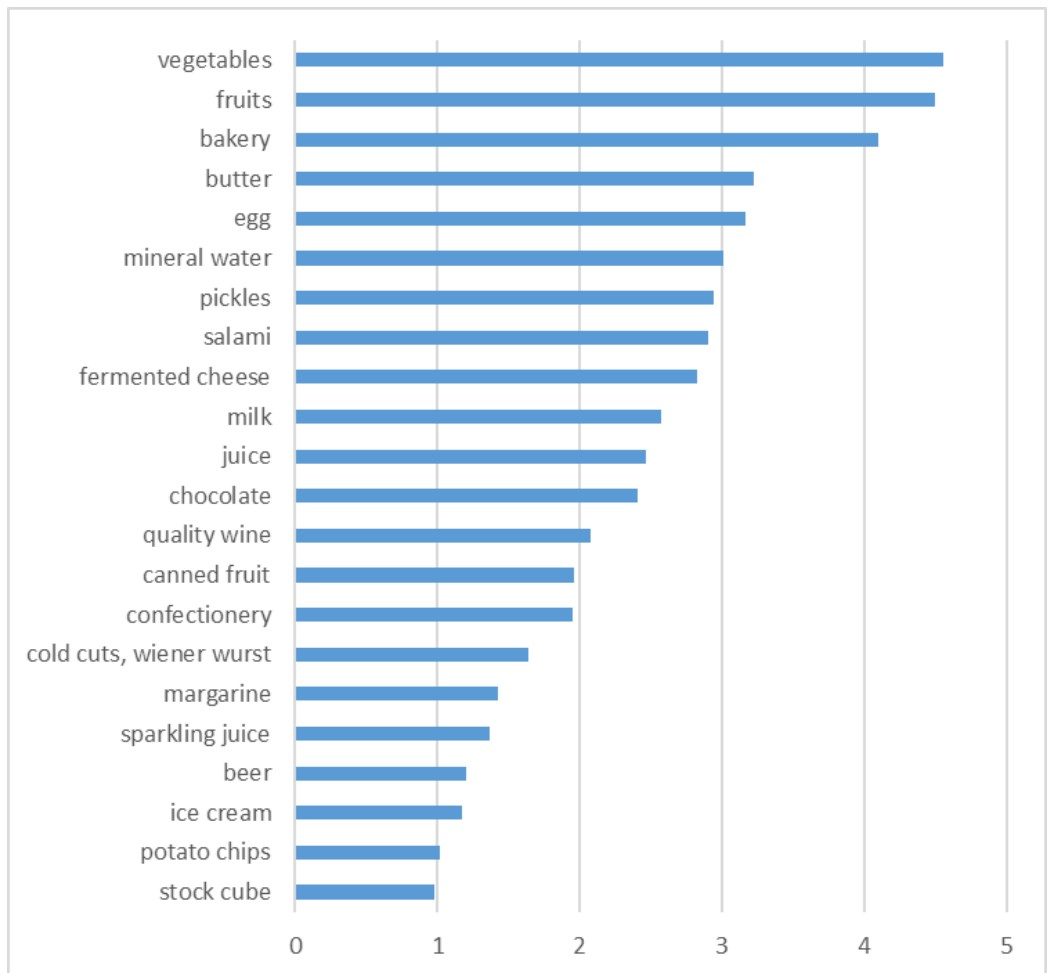

**Figure 4.** Consumption frequency of certain food products in the case of organic market consumers (N = 102) *Source: Own data collection and processing, 2018.*

The organic food consumers buy food with the highest frequency on the traditional market which is followed by the direct purchase from the producers and the local producers' market. This finding is in accordance with other relevant surveys. The standard deviation was the lowest in the case of the organic market, while it was the highest in the case of direct relations with the producers.

Another interesting question is where organic food buyers purchase organic food products. We found on the basis of our survey that they prefer the organic markets which were followed by special organic stores and retail chains.

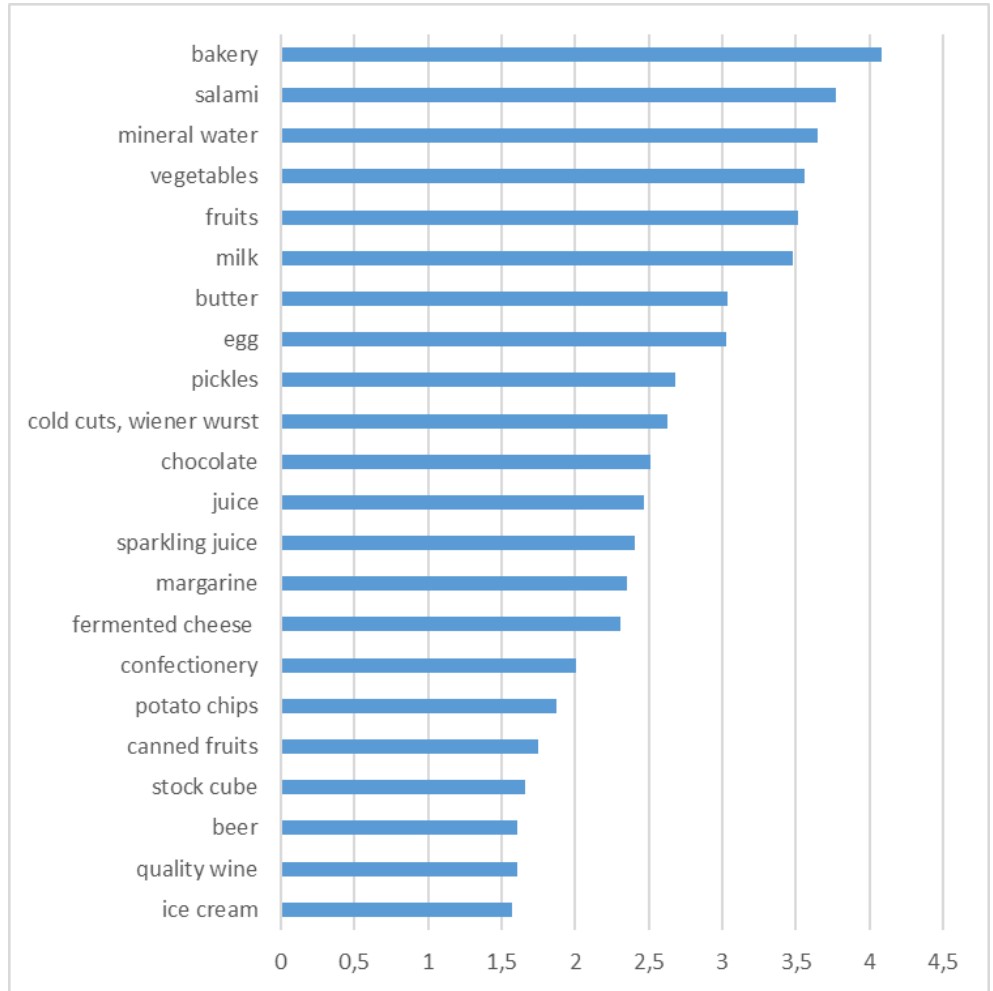

**Figure 5.** Consumption frequency of certain food products in the case of non-organic food consumers (N = 582) *Source: Own data collection and processing, 2018.*

*4.2. The Outcomes Of Factor Analysis*

The factor analysis resulted in 18 observed variables allocated to six factors. "Color" as an organoleptic characteristic is a separate explaining variable. It is interesting, because we noticed during the interviews on the Organic Market, that most of the organic food consumers think that color is not as important as taste or freshness, so they did not link color with freshness. However, they usually buy fruits and vegetables in the organic market. The six factors were named "Health effect", where the information on label plays an important role, "Influences", "Organoleptic properties", Convenience", "Price and habits", where this assigned quality characteristic and the habits have influence (Table 3).

**Table 3.** Results of the factor analysis.

| | Components | | | | | |
| --- | --- | --- | --- | --- | --- | --- |
| | **Health Effect** | **Influences** | **Organoleptic Properties** | **Convenience** | **Price and Habit** | **Colour** |
| Price | −0.019 | −0.014 | −0.065 | 0.132 | **0.892** | −0.020 |
| Wrapping | 0.012 | **0.437** | −0.404 | 0.433 | 0.053 | 0.138 |
| High endurance | 0.074 | 0.292 | 0.038 | **0.691** | 0.133 | −0.256 |
| Advertisement | −0.067 | **0.740** | 0.073 | 0.193 | −0.041 | 0.242 |
| Origin | **0.624** | 0.298 | 0.031 | −0.175 | 0.052 | 0.029 |
| Brand | 0.073 | **0.793** | 0.158 | 0.040 | 0.013 | 0.037 |
| Label | **0.514** | 0.385 | −0.025 | −0.232 | −0.230 | −0.178 |
| Contents | **0.710** | −0.045 | 0.100 | 0.112 | −0.078 | −0.157 |
| Additives | **0.772** | −0.199 | −0.034 | −0.059 | −0.150 | 0.190 |
| High preparedness | −0.257 | −0.107 | 0.319 | **0.679** | 0.015 | 0.183 |
| Nutrients | **0.669** | −0.054 | −0.001 | 0.247 | 0.135 | 0.184 |
| Free of E- numbers | **0.814** | −0.083 | −0.065 | −0.109 | −0.048 | 0.137 |
| Taste | 0.047 | 0.124 | **0.719** | 0.140 | 0.107 | 0.330 |
| Colour | 0.117 | 0.255 | 0.131 | −0.036 | −0.013 | **0.843** |
| Freshness | 0.187 | 0.146 | **0.759** | 0.130 | −0.139 | −0.069 |
| Recommendation | −0.055 | **0.591** | 0.497 | −0.075 | 0.272 | 0.029 |
| Habit | −0.288 | 0.153 | 0.453 | −0.046 | **0.506** | 0.028 |
| Good impact on health | **0.656** | 0.134 | 0.099 | −0.092 | −0.060 | −0.101 |

Extraction Method: Principal Component Analysis. Rotation Method: Varimax with Kaiser Normalization (Rotation converged in 6 iterations).

Finally, 4 clusters could be separated according to the abovementioned on the basis of the factor analysis. The most important characteristic of members of Cluster 1 (N = 59) (Figure 6) that the price of organic food influences them to the least extent as compared to the other segments' members. The freshness and good health impact of food are very important for them. They seem very health-conscious consumers, who are inclined to pay higher prices for quality and healthy food. They try to buy as much healthier food as they can (Health conscious). Freshness and a good impact on health are the most important motives for them. Price and advertisement have the least influence on their purchase decisions. They also try to buy as healthy food as they can and think the food is full of harmful components. As a consequence of this, they select carefully the source of purchase and they do not trust in food sold in food stores and judge poorly the performance of the authority. Freshness, taste, and food components are the most important motives in the second segment of the consumers, Cluster 2 (N = 88) (Figure 6). They chose organic food because they are disappointed by conventional food. They pay much attention to where they buy food and think that food is full of harmful components. These are the most influencing factor in their attitude (Disappointed). Members in Cluster 3 (N = 21) (Figure 6) choose organic food because of its favorable health effect, but they pay attention to the price also. They have a fear of additives, so the most influencing motive for them is that the food should not contain additives (Safe and free food eaters). They do not trust in food sold in the food stores and judge poorly the performance of the authority as the members of Cluster 1.

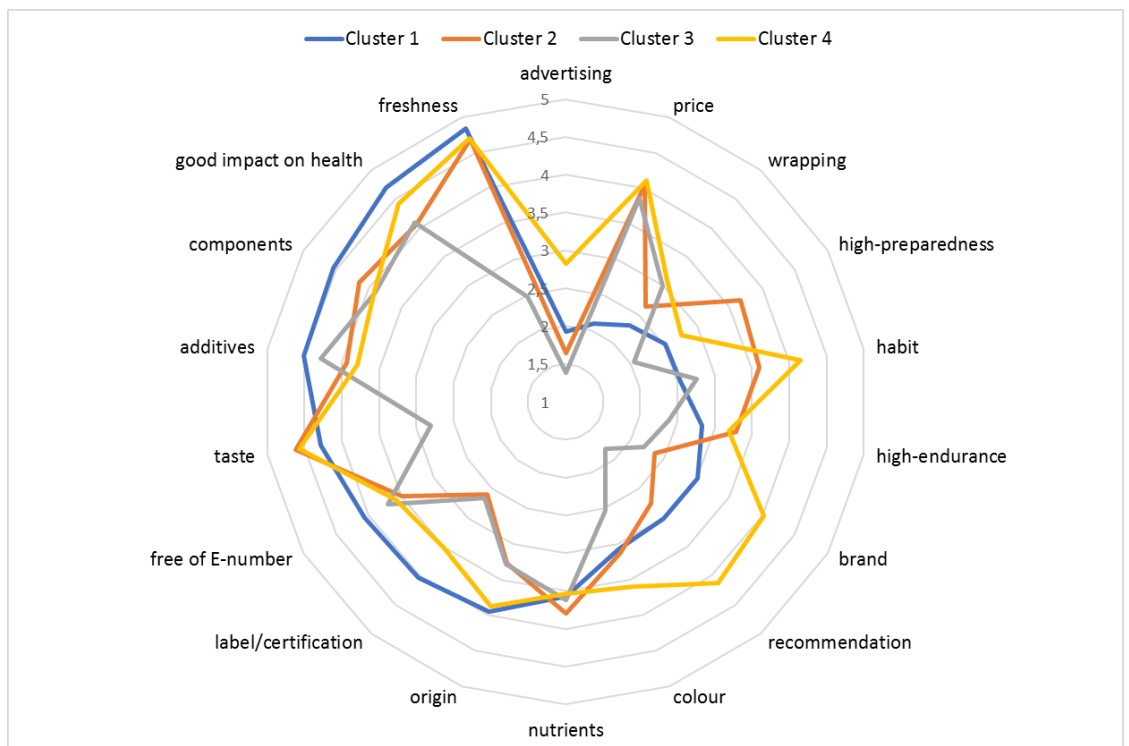

**Figure 6.** Factors influencing consumers' decisions in different clusters.

Cluster 4 (N = 78) (Figure 6) also likes eating well, and think organic food is tastier and fresher than conventional but they can be influenced and have stable purchasing source. Since they influenced by habits in purchasing the brand as a motive is ranked to the best place in this cluster as compared to the other segments. They are lag behind in trying new products (Quality – price comparator). The attitude "Food is full of harmful components" is ranked in the first two places in every cluster, which also very important information for policymakers and other stakeholders in the food industry. It is important to note that "label" as a motivator is ranked in the middle or end of the motivation list in every cluster which focuses our attention on the role of labels in the promotion of organic products.

## 5. Discussion and Conclusions

Our research has confirmed the connection found in the literature that consumers mainly buy fresh food on the bio market, which mainly means buying fruits and vegetables [12]. We also see through consumer perception of labels and product labels, because of the microbiological risks associated with the consumption of fresh foods, consumers are more careful when purchasing [13,14]. From the literature review, it can be concluded that consumers of organic food can be divided into several groups based on the main behavioral categories. According to the literature, the two most influential groups are the group that prefers healthy food and the group that follows fashion or current trends [53,54,58]. The preference for freshness of products (which in many cases is also a symbol of healthy food) is an important criterion not only for organic food consumers but also for traditional food consumers. According to our survey, it can be stated that organic food consumers have the healthiest diet because they eat vegetables and fruits with the highest frequency. Organic market consumers pay much attention to where they buy food, but the respondent of the survey does not try to buy food from small scale farmers directly it is not important for them. They do not really think that they get safer food on the market and they do not trust in food traded by food stores, the most trust in their own senses. The surveyed consumers decide mainly on the basis of sensory characteristics (freshness, flavor) but the positive health impact also a very dominant factor at food purchase. According to the factor analysis we can state that there are six explaining factors, we named them as "Health effect", where the

information on label plays an important role, "Influences", "Organoleptic properties", Convenience", "Price and habits", where this assigned quality characteristic and the habits have influence. "Colour" as an organoleptic characteristic is a separate explaining variable. Four clusters with different purchasing motives could be separated and some differences can be found among the clusters in attitude toward food purchase. We named these clusters as Health conscious, Disappointed, Safe and free food eaters, Quality-price comparator. Freshness and good impacts on health are the most important motives for the members of the "Health-conscious" cluster. Price and advertisement have the least influence on their purchase decisions. Freshness, taste, and food components are the most important motives in the segment of the "Disappointed" consumers. "Safe and free food eaters" have a fear of additives, so the most influencing motive for them is that the food should not contain additives. They do not trust in food sold in food stores and judge poorly the performance of the authority as the members of cluster 1. "Quality-price comparator" think organic food is tastier and fresher than conventional but they can be influenced and have stable purchasing source. The brand as a motive is ranked to better place in this cluster as compared to the other segments. The attitude "Food is full of harmful components" is ranked in the first two places in every cluster, which is very important information for policymakers and other stakeholders in the food industry. It is important to note that "label" as a motive is ranked in the middle or end of the motive list in every cluster which focuses our attention on the role of labels in the promotion of bio-foods and also for the malfunctions of these labels. The ingredients should be well readable and markable on labels, however the origin and brand are almost irrelevant for the consumers' segments. The presence of certain ingredients in the product and the freshness of the products can contribute more to sales than branding. Based on the results of the study, we can call the group of health-conscious consumers a clearly distinct group. In their case, it is a surprising result that this group can be influenced well by different marketing tools towards healthy eating. It is an interesting finding for disappointed consumers that they try to get back the loss of consumer confidence in traditional products through their preference for the taste of the products. Consumer sensitivity to the price of the product is typical of the quality-price comparator group, but it is interesting to observe from studies that consumers pay close attention to the price of products in almost all groups. Brand awareness and personal recommendations are of paramount importance to the Safe and Free Food Eaters group. The price comparison consumer group is that they do not attach a particularly strong preference to either the product characteristics or the conditions under which the product is sold at the time of purchase. The results of the study provide valuable and important information about Hungarian consumers of organic food products that can be used by decision-makers in their development strategies in order to enhance small farmers' production and local products. Producers should accommodate their supply to the special need of the consumers' preferences to develop their products based on the needs of each group of consumers. The level of consumers' satisfaction can be increased and a profitable production can evolve. It is important to note that consumers generally believe that "Food is full of harmful ingredients" - this is the first two places in each cluster, which is very important information for policymakers and food business operators. It is important to emphasize that the "label" as a motivator is ranked in the middle or at the end of the motivation list in each group, which is important to consider because we previously thought based on the related researches, that good brands and well-known branding can create an effective sales strategy. Based on the feedback we can confidently state that the information on the label is really important, but branding itself is only a secondary consumer demand!

**Limitations:** The research was carried out on a small sample, so no exact behavioral patterns can be deduced from the composition of the respondents. It should be emphasized that the data were processed from two sources. In a sample of 31 people, we conducted a bio-market survey among bio-market buyers and interviewed over 800 people with university students to learn about family consumption patterns. Our study does not cover all possible topics but is limited to what we consider important. The research carried out does not represent the consumer habits of all organic food consumers living in Hungary, but the results identify appropriate trends in the consumer community.

**Author Contributions:** Conceptualization, methodology and formal analysis, K.N.-P.; resources, writing, writing—review and editing, and supervision, C.F.

**Funding:** This research received no external funding.

**Acknowledgments:** Preparation of the manuscript and our final article was supported by the Szent Istvan University (SZIU) Climate Change Research Centre and Doctoral School of Management and Business Administration at SZIU. Special thanks to Amelia Godor, Miriam Bahna, and Prespa Ymeri PhD students who helped us with the questionnaire management and the collection and sorting of references.

**Conflicts of Interest:** The authors declare no conflict of interest.

## Appendix A

The applied questionnaire

1. How often do you consume the following food categories?

   0—never, 1—less than once in a month, 2—1–2 times in a month, 3—1–2 times in a week, 4—3–4 times in a week, 5 every day

| | | | | | | |
|---|---|---|---|---|---|---|
| **stock-cube** | 0 | 1 | 2 | 3 | 4 | 5 |
| **ice cream** | 0 | 1 | 2 | 3 | 4 | 5 |
| **potato chips** | 0 | 1 | 2 | 3 | 4 | 5 |
| **beer** | 0 | 1 | 2 | 3 | 4 | 5 |
| **sparkling juice** | 0 | 1 | 2 | 3 | 4 | 5 |
| **canned fruit** | 0 | 1 | 2 | 3 | 4 | 5 |
| **quality wine** | 0 | 1 | 2 | 3 | 4 | 5 |
| **confectionery** | 0 | 1 | 2 | 3 | 4 | 5 |
| **cold cuts, wiener wurst** | 0 | 1 | 2 | 3 | 4 | 5 |
| **margarine** | 0 | 1 | 2 | 3 | 4 | 5 |
| **juice** | 0 | 1 | 2 | 3 | 4 | 5 |
| **chocolate** | 0 | 1 | 2 | 3 | 4 | 5 |
| **fermented cheese** | 0 | 1 | 2 | 3 | 4 | 5 |
| **pickles** | 0 | 1 | 2 | 3 | 4 | 5 |
| **milk** | 0 | 1 | 2 | 3 | 4 | 5 |
| **egg** | 0 | 1 | 2 | 3 | 4 | 5 |
| **butter** | 0 | 1 | 2 | 3 | 4 | 5 |
| **salami** | 0 | 1 | 2 | 3 | 4 | 5 |
| **mineral water** | 0 | 1 | 2 | 3 | 4 | 5 |
| **fruits** | 0 | 1 | 2 | 3 | 4 | 5 |
| **bakery** | 0 | 1 | 2 | 3 | 4 | 5 |
| **vegetables** | 0 | 1 | 2 | 3 | 4 | 5 |

2. How are you affected by the following factors in your food purchase? Please, indicate it with 1–5.

   1—I am not affected at all, 5—I am affected in a great extent

| |
|---|
| advertising |
| high-preparedness |
| wrapping |
| high-endurance |
| price |
| habit |
| colour |
| label/certification |
| nutrients |
| recommendation |
| free of E-number |
| brand |
| additives |
| origin |
| components |
| good impact on health |
| taste |
| freshness |

3.  What do you think about the health impact of the following articles?

    1—it is not harmful, 5—very dangerous

| |
|---|
| animal fat |
| exposing food to smoke |
| salt |
| high energy content |
| high fat content |
| sweetening content |
| allergens |
| trans-fatty acid |
| additives |
| high sugar content |
| mold |
| preservatives |
| GMO content |
| artificial food colouring |
| antibiotic residue |
| pesticide residues |
| hormone residues |
| advertisement |
| wrapping |
| easy to prepare |
| brand |
| long shelf life |
| recommendation |
| color |
| label |
| routine |
| price |
| origin |
| nutrient content |
| free from E-number |
| free from additives |
| ingredients |
| positive impacts to health |
| flavour |
| freshness |

4.  Do you buy any food direct from the producer?

    1—Yes

    2—No

    If yes, list it, please!

5.  How often do you use the following supply channels?

    Please, use the following code!

    0—never, 1—less than once in a month, 2—1–2 times in a month, 3—1–2 times in a week, 4—3–4 times in a week, 5 every day

| |
|---|
| local producers' market |
| organic market |
| directly from producers |
| through the internet |
| traditional market |
| other direct selling form |

6. How often do you buy organic food?

   0—never, 1—less than once in a month, 2—1–2 times in a month, 3—1–2 times in a week, 4—3–4 times in a week, 5 every day

7. Where you usually buy organic food?

   organic market
   directly from producers
   through the internet
   special organic food store
   retail chain
   other:________________________

8. Are you up-to-date in food-safety issues? Please, indicate it with 1–10. 1—I am totally uninformed, 10—I am very familiar with the topic.

9. What is your opinion about the food-safety situation in Hungary? Has it changed in negative or positive direction in the last 10 years? 1—it has been getting worse in a great extent, 2—it has been getting worse, 3—it has not changed, 4—It has improved in some extent, 5—It has improved much

10. Are you agree with the following statements? Please, indicate with 1–5!

    1—I am totally disagree with it, 5—I am totally agree with it.

    I trust in food sold by food stores.
    I found the activity of the authorities adequate.
    I try to buy food made by small scale farmers.
    I always read the name of the processor on the label.
    I can buy safer food on the market.
    I buy only trusty food.
    I prefer Hungarian food.
    I try to purchase healthy food.
    Most of the food is full of harmful ingredients.
    I take care of that where I buy food.
    I think I am a conscious consumer.

11. What is your gender?

    [] female

    [] male

12. What is your highest level of education? Please, indicate it!

    [] primary school

    [] vocational school

    [] technical college

    [] grammar school

    [] college/university

13. The age of the respondent:

    [] under 20 [] between 20 and 30 [] between 31 and 40 [] between 41 and 50 [] between 51 and 65 [] more than 65

14. Settlement type of the respondents' residence:

    [] Budapest (capital city) [] chief town of a county [] other city [] village [] ranch

15.　　Occupation of the respondent:

　　　[] unskilled worker [] skilled worker [] entrepreneur [] employee [] manager [] pensioner [] student

16.　　Net income in your family

　　　much smaller than the average

　　　smaller than the average

　　　average

　　　more than the average

　　　much more than the average

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
