# Peer review of "Important Influencing and Decision Factors in Organic Food Purchasing in Hungary"

_sustainability, doi:10.3390/su11216075_

Round 1

Reviewer 1 Report

Dear Authors,

I think this paper has a certain potential to contribute to Sustainability but there are some major improvements needed:

The title does not match the content of the manuscript. The abstract is clumsily written and should be revised to clearly state the aim of the study, methods applied and provide a concise summary of main findings; What do mean by “The organic food consumption is the most developed and widely accepted producing form?” “We used an arbitrary sample of 247 questionnaires?” Line 97 – “Szente et al (2015) mentioned that in Hungary several times not necessary products are distributed, while the selection and volume of certain products are not satisfactory on organic markets [21]”. Correct the reference and make it clear what you mean by not necessary products? Please make a clear distinction in the literature review between different labels/logos for organic food (EU, private labels etc) and brands. Make it explicit what is the role of different labels and brands for organic food consumers. Reflect on it how the way of labelling influence their purchasing decisions? Line 102 “Rácz found that domestic consumers are not know the objective meaning of food labels in several cases...” year? Not know to? Line 149 unclear “Due to the above-mentioned phenomena, it can be stated that organic food consumers are not very interested in branded products, but rather they are looking for product groups or opting for system characteristics related to product sales that are based on consumer confidence” Line 152 please make it explicit what are the main aims of the study Line 155 Material and methods – are not structured in a logic way to provide a comprehensive overview of the data collection and analysis. Sampling and survey instrument – please make a clear overview of the sampling methods and procedures including a graph or any other way of visualisation. What did you learn from FGI? How did you use this results to improve the questionnaire? Line 175-176 please provide the references you used to develop the questionnaire The structure of the questionnaires should be better described. Line 181-182 correct should be Likert not Lickert but this is not Likert scale. Line 187-190, please provide any references Line 213 Data analysis – please add details on SPSS – publisher, describe more in detail the segmentation process. How did you perform factor analysis? How did you come to optimal number of clusters? Keep in mind the limitations of k-means. How did you compare means? Please add the indexes to show the differences between clusters. Line 230 – please include the profile of the sample and the clusters identified in a table Figure 1. What does it refer to, it is primary or secondary data? Why you decided to use it in the RESULTS if it comes from the publication of “TörĹ‘csik, M.; Pál E.; Jakopánecz, E. (2018): New and innovative consumer demands and expectations on the Hungarian food market . – In: Acta Agraria Debreceniensis, ISSN 1587-1282 , 2018. Spec edition, 485-499. p. 7”? The scale on Figure 1 is not correct. The same remark for other figures in the manuscript. Please add an introductory paragraph to describe what you really want to present in the result section – as such it is totally confusing. Figure 2 – I do not really understand what kind of data you present? “It is important and on the contrary to the average Hungarian consumers behaviour that the price is not really important for this segment (Figure 2)? How did you come to this conclusion? Figure 3 and 4 try to combine it and present in a more structured way Line 293 – details of the factor analysis should be in the material and methods Instead of Figures 5-8 use the table and make it clear what are the differences between segments using indexes. - Discussion of results is missing. The conclusion are poor and do not go beyond a simple summary of results. More in depth reflection on the research findings would be appreciated. The bibliography, tables, figures are not in line with Sustainability requirements.

Author Response

Dear Reviewer (1),

Thank you for your comments, which have greatly helped to improve the quality of the paper. All comments and suggestions have been accepted and incorporated into the revised version. Here are the answers to your comments:

„The title does not match the content of the manuscript.”

The title has been adapted to the content.

„The abstract is clumsily written and should be revised to clearly state the aim of the study, methods applied and provide a concise summary of main findings;”

It has been corrected.

 „What do mean by “The organic food consumption is the most developed and widely accepted producing form?”

In terms of healthy food production and sustainability, it is the most accepted production system based on consumer opinion.

“We used an arbitrary sample of 247 questionnaires?”

The selection of the sample was arbitrary, but the process was described in detail. The word (arbitrary) has been deleted.

Line 97 – “Szente et al (2015) mentioned that in Hungary several times not necessary products are distributed, while the selection and volume of certain products are not satisfactory on organic markets [21]”. Correct the reference and make it clear what you mean by not necessary products?

Not looking for import products. The range is widening, but there is no demand for them. Redrafted.

„Please make a clear distinction in the literature review between different labels/logos for organic food (EU, private labels etc) and brands. Make it explicit what is the role of different labels and brands for organic food consumers. Reflect on it how the way of labelling influence their purchasing decisions? Line 102 “Rácz found that domestic consumers are not know the objective meaning of food labels in several cases...” year? Not know to?”

We have restructured this section of the literature review. Questions that could not be satisfactorily answered in the present investigation have been removed. According to Rácz (2013), “consumers are not able to interpret different labels, either the EU or private labels”.

"Line 149 unclear - “Due to the above-mentioned phenomena, it can be stated that organic food consumers are not very interested in branded products, but rather they are looking for product groups or opting for system characteristics related to product sales that are based on consumer confidence”

Consumers prefer direct producer relationships and traditional forms of marketing to organic food purchases due to the variety of labels, the lack of clarity and the lack of trust.

"Line 152 – „please make it explicit what are the main aims of the study „

It was formulated in the abstract and at the beginning of the introduction. - „What are the main influencing factors of purchase decisions?”

"Line 155 - Material and methods – are not structured in a logic way to provide a comprehensive overview of the data collection and analysis. Sampling and survey instrument – please make a clear overview of the sampling methods and procedures including a graph or any other way of visualisation.

The selection of the sample was arbitrary and its process was described in detail. Due to size limitations, we did not attach a chart.

What did you learn from FGI? How did you use this results to improve the questionnaire? Line 175-176 please provide the references you used to develop the questionnaire. The structure of the questionnaires should be better described.

We attached the questionnare and we have only few modification affecting mainly the editing and some missing information.

"Line 181-182 correct should be Likert not Lickert but this is not Likert scale.  

It has been corrected.

"Line 187-190, please provide any references

There is no reference, consumers mentioned in our survey of the organic market. We removed this sentence.

"Line 213 Data analysis – please add details on SPSS – publisher, describe more in detail the segmentation process. How did you perform factor analysis? How did you come to optimal number of clusters? Keep in mind the limitations of k-means. How did you compare means? Please add the indexes to show the differences between clusters.

We used the latest version available on the IBM website (monthly subscription). Varimax rotation was used in the factor analysis. Groups were formed using hierarchical clustering, and the number of groups was determined using the elbow method. The required table is inserted.

"Line 230 – please include the profile of the sample and the clusters identified in a table Figure 1. What does it refer to, it is primary or secondary data? Why you decided to use it in the RESULTS if it comes from the publication of “TörĹ‘csik, M.; Pál E.; Jakopánecz, E. (2018): New and innovative consumer demands and expectations on the Hungarian food market . – In: Acta Agraria Debreceniensis, ISSN 1587-1282 , 2018. Spec edition, 485-499. p. 7”?

These are primary data, the reference was confusing and therefore removed, not required.

"The scale on Figure 1 is not correct. The same remark for other figures in the manuscript. Please add an introductory paragraph to describe what you really want to present in the result section – as such it is totally confusing.2

The scale has been corrected.

"Figure 2 – I do not really understand what kind of data you present? “It is important and on the contrary to the average Hungarian consumers behaviour that the price is not really important for this segment (Figure 2)? How did you come to this conclusion? Figure 3 and 4 try to combine it and present in a more structured way"

This is data for our entire sample, which can be broken down into three groups. For organic market buyers, organic consumers and the non-consumer group. Those who consume organic products but are not eco-market consumers do not go to the eco-market and buy elsewhere. The needed information and explanation were added.

"Line 293 – details of the factor analysis should be in the material and methods Instead of Figures 5-8 use the table and make it clear what are the differences between segments using indexes. –

A table showing the differences between the clusters was inserted and the figures were merged into a figure.

"Discussion of results is missing.

We interpreted the results in the Conclusions section.

"The conclusion are poor and do not go beyond a simple summary of results. More in depth reflection on the research findings would be appreciated.

The Conclusions chapter has been reworded.

"The bibliography, tables, figures are not in line with Sustainability requirements.

Have been corrected.

Thank you very much for your suggestions and comments. The quality of the article has improved significantly based on your reviewer opinion!

Authors

Reviewer 2 Report

General comment:

The authors used a principal component analysis for segmentation of a group of ‘organic’ consumers. This part is interesting. However the paper is not concise and not to the point. Some statements are used without proper reference. The English needs extensive editing. The order is not logical. Some parts are very unclear.

Specific comments:

Abstract

L14 what is meant by ‘families with little child or children’ ? Do you mean small children or few children?

Introduction is very wordy, and not at all to the point.

L54 ‘on the basis of our survey’ it is unclear which survey is meant

Literature research:

This part is very difficult to read. It is not well structured. Subheadings could improve it. The relationship to the research questions and the study is unclear. It would be more logical to discuss the results in the light of this literature review. In addition some parts fit better with the introduction: such as mentioning of the Hungarian organic market size.

L96-98: Unclear what is meant by this sentence.

L119-120: this should be referenced.

Methods:

It is unclear what is done, and which data are used.

L158-170: Why are the 31 questionnaires being collected ? For what purpose? For an inventory of good questions?

L172: a questionnaire is applied: can you refer to it? (in English)

L187-190: why this text? If this is part of your research question, it should be mentioned in the introduction.

L193: After the above mentioned on the spot survey…which one? How many participants?

L194: a further 811 were collected: what & how?

Did you have participants in this study or questionnaires?

An ethical statement is missing.

Results:

L224-228: Good to mention this is a summary of the results.

L225 mentiones ‘unlike traditional consumers’ but these were not part of your study. So this cannot be part of the results. It should be part of a discussion.

Last sentence: L227-228: what do you mean?

L248 ‘extrinsic attributes’ can you mention any examples?

Figure 2: scale is not a likert-scale

L261: mention of non-organic consumers, these were not included? How do you know?

Tables 1 and 2 need more explanation: why are they there?

L293-296: move to method section.

L310-352: these clusters are the most interesting part of the study. The text in the conclusions (L372-376) should be part of the results.

Author Response

Dear Reviewer (2),

Thank you for your comments, which have greatly helped to improve the quality of the paper. All comments and suggestions have been accepted and incorporated into the revised version. Here are the answers to your comments:

General comment:

„The authors used a principal component analysis for segmentation of a group of ‘organic’ consumers. This part is interesting. However the paper is not concise and not to the point. Some statements are used without proper reference. The English needs extensive editing. The order is not logical. Some parts are very unclear.”

We have redesigned the article and added a new logic to the content. We avoid unnecessary tables and supplement the introduction with resources that support the hypotheses.

Specific comments:

Abstract

L14 what is meant by ‘families with little child or children’ ? Do you mean small children or few children?

„We meant small children” – has been corrected!

„Introduction is very wordy, and not at all to the point.”

We systematized the introduction and presented the trends and impacts that are currently affecting the demand for organic food.

L54 ‘on the basis of our survey’ it is unclear which survey is meant

Has been deleted this sentence!

Literature research:

„This part is very difficult to read. It is not well structured. Subheadings could improve it. The relationship to the research questions and the study is unclear. It would be more logical to discuss the results in the light of this literature review. In addition some parts fit better with the introduction: such as mentioning of the Hungarian organic market size.”

The literature review is organized around 3 topics that are relevant and topical to the Hungarian eco-market. The introduced topics were processed with a large number of references.

L96-98: Unclear what is meant by this sentence.

It has been changed!

L119-120: this should be referenced.

Corrected with references!

Methods:

„It is unclear what is done, and which data are used.”

The sampling process has been described and the details have been compiled in a well-tracked manner.

L158-170: Why are the 31 questionnaires being collected ? For what purpose? For an inventory of good questions?

One of the important aspects was to check the accuracy of the questionnaire and to collect data. We also obtained a lot of other information through personal interviews.

L172: a questionnaire is applied: can you refer to it? (in English)

Attached into the Appendix!

L187-190: why this text? If this is part of your research question, it should be mentioned in the introduction.

Has been deleted from the text.

L193: After the above mentioned on the spot survey…which one? How many participants?

There were 31 respondents in the organic market.

L194: a further 811 were collected: what & how?

Did you have participants in this study or questionnaires?

As described, the questionnaires were collected with the help of university students.

An ethical statement is missing.

Has been added!

Results:

L224-228: Good to mention this is a summary of the results.

We did not use the Discussion section, so we have summarized the details here.

L225 mentioned ‘unlike traditional consumers’ but these were not part of your study. So this cannot be part of the results. It should be part of a discussion.

This was part of the result, as 582 of the total sample do not consume organic products. Missing information was replaced in the text.

Last sentence: L227-228: what do you mean?

Some national expressions do not follow international trends

L248 ‘extrinsic attributes’ can you mention any examples?

Like colour and size or wrapping.

Figure 2: scale is not a Likert-scale            

Has been corrected.

L261: mention of non-organic consumers, these were not included? How do you know?

This is part of the sample, has been included!

Tables 1 and 2 need more explanation: why are they there?

Has been added more explanation!

L293-296: move to method section.

Has been moved over!

L310-352: these clusters are the most interesting part of the study. The text in the conclusions (L372-376) should be part of the results.

The chapter has been reviewed and edited. Thanks for the comment.

Thank you very much for your suggestions and comments. The quality of the article has improved significantly based on your reviewer's opinion!

Authors

Reviewer 3 Report

Dear authors,

Your article approaches a topic of interest because is related to organic food. In order to substantially improve your paper I could give you some advices:

Section 2- needs your attention and it must be rewritten. I would advice to divide the information in 2-3 subsections and choose a title for them (e.g. 2.1. organic food consumption in Hungary; 2.2. Importance of label etc.). 

Beside that, supplementary references must be included, mostly because there is a lot of research on this topic and your reference list is quite poor.

There are many sentences which need your attention. I am not an English native speaker, but when I read your paper I noticed that the language and phrasing is at a medium level, so I advice that a professional speaker to check your paper.

4. Results

A table with socio-demographic data is more suitable

Table 2 must present final results, where factors ca become obvious.

Results must be presented in the context of other researches. No mentions of other similar researches were found.

refereneces must be written according to the journals' requirements

Author Response

Dear Reviewer (3),

Thank you for your comments, which have greatly helped to improve the quality of the paper. All comments and suggestions have been accepted and incorporated into the revised version. Here are the answers to your comments:

„Your article approaches a topic of interest because is related to organic food. In order to substantially improve your paper I could give you some advices:

Section 2- needs your attention and it must be rewritten. I would advice to divide the information in 2-3 subsections and choose a title for them (e.g. 2.1. organic food consumption in Hungary; 2.2. Importance of label etc.). „

We modified it based on the suggestion.

„Beside that, supplementary references must be included, mostly because there is a lot of research on this topic and your reference list is quite poor.”

New references (15) have been added.

„There are many sentences which need your attention. I am not an English native speaker, but when I read your paper I noticed that the language and phrasing is at a medium level, so I advice that a professional speaker to check your paper.”

We checked it out with a native English-speaking researcher.

„4. Results”

„A table with socio-demographic data is more suitable”

Socio-demographic data provided no additional information.

„Table 2 must present final results, where factors ca become obvious.”

In other publications this is the form, and we followed the publication practice.

„Results must be presented in the context of other researches. No mentions of other similar researches were found.”

In the bibliography, we have referred to similar publications.

„refereneces must be written according to the journals' requirements”

Have been corrected.

Thank you very much for your suggestions and comments. The quality of the article has improved significantly based on your reviewer opinion!

Authors

Round 2

Reviewer 1 Report

Dear Authors,

I appreciate your efforts but the  manuscript still needs quite some improvements. My major concern is unsatisfactory presentation of the cluster analysis results and lack of any discussion of your results with the findings in the literature. English language must be verified by an expert in the field, preferably native speaker. 

Some other comments:

Line 189 - A focus group consisting of three 190 academics experts was formed to ensure the validity and suitability of the items.  – three experts can be considered a focus group.

The limitation of the study resulting from combining data from different sources should be clearly stated and further discussed.

The questionnaire was established mainly on the basis of the relevant Hungarian professional literature selected according to the aim of the article – any references?

The figures are still not correct – Liker scale is 1-5 not 0-5, remove the “source” – it is obvious it is your own data, correct the graphs to ensure consistency throughout the manuscript

I asked in my previous review to describe how you tested the differences among clusters. The figure 7 you added is not correct – wrong scale. It is essential to provide the information on the statistically significant differences between clusters.

Discussion is still missing – there is no reference to literature review.

Conclusion section should not be used for the interpretation of the results as Authors claim. Any conclusions should be drawn.

Tables – must be in line with Sustainability template

Author Response

Dear Reviewer1,

All comments and additional suggestions have been accepted from your review. The points given have been checked and corrected. At the end of the results section, we are able to construct a very illustrative graph that can easily help us to see the research results for each cluster. Corrected text is highlighted in red throughout the text so changes can be easily tracked.
Thank you very much for your hard work and we think we can improve our professional knowledge based on your suggestions.

the Authors

Reviewer 3 Report

Dear authors,

Your work is substantially improved.

Good luck

Author Response

Dear Reviewer3, 

Thank you for your positive comments and good luck wishes!

the Authors

Round 3

Reviewer 1 Report

Dear Authors,

I asked in two of my previous review to describe how you tested the differences among clusters. It is essential to provide the information on the statistically significant differences between clusters to discuss the results !

Discussion is still missing – there is no reference to results of any other researchers !

Tables, figures – must be in line with Sustainability template. It is not acceptable how it is not. Particularly Table 1!

Author Response

Dear Reviewer1,

Thank you very much for your comments. We made the following changes to this article:

Based on Table 1, further tests and parameters (eg demographics) can be used to confirm the goodness of the clusters, but this will be investigated in the next phase.

The discussion and conclusion sections were based on the Publons Academi Guide, and we responded accordingly to this section:

â—Ź Are the results discussed from multiple angles and placed into context without being overinterpreted?
â—Ź Do the conclusions answer the aims of the study?
â—Ź Are conclusions supported by references or results?
â—Ź Are the limitations of the study fatal or are they
opportunities to inform future research?

The tables and figures have been modified!

Thanks,
the Authors